# Regulation of *MdANR* in Anti-Burning Process of Apple Peel

**DOI:** 10.3390/ijms26104656

**Published:** 2025-05-13

**Authors:** Yifeng Feng, Wenya Tian, Junjiao Guo, Jianghong Fu, Jiangbo Wang, Yan Wang, Zhengyang Zhao

**Affiliations:** 1College of Horticulture and Forestry, Tarim University, Alar 843300, China; 2College of Horticulture, Northwest A&F University, Yangling 712100, China

**Keywords:** *MdANR*, regulate, apple fruit, resist sunburn

## Abstract

Sunburn in apple peel significantly affects fruit appearance and reduces its commercial value. Previous research has shown that apple peel reduces sunburn by increasing the accumulation of proanthocyanidins (PAs) and other protective compounds. However, the precise molecular regulatory mechanism remains unclear. In this study, we systematically investigated *MdANR*, a key gene involved in PAs biosynthesis. We found that *MdANR* expression in apple peel is responsive to temperature and light fluctuations, with higher expression levels observed under increased temperature and light exposure. Functional analysis revealed that *MdANR* overexpression in apple peel and callus enhanced resistance to high-temperature and -light-intensity stress, accompanied by a corresponding increase in PAs and chlorogenic acid contents. In addition, we demonstrated that MdMYBR9 can activate *MdANR* promoter activity and promote its expression through yeast one-hybrid, dual-luciferase, and electrophoretic mobility transfer experiments. The results indicated that *MdMYBR9* was an upstream regulator of *MdANR*. Based on these findings, this study proposes the *MdMYBR9-MdANR-PAs* regulatory model for apple sunburn resistance, providing a molecular framework for enhancing sunburn tolerance in apple breeding programs.

## 1. Introduction

Sunburn is a common physiological disorder that significantly reduces fruit yield and is a key factor limiting fruit production in summer. It occurs when the temperature and light intensity on the fruit surface exceed a critical threshold [1]. Studies have documented the effect of sunburn in various tree species, including apples [2,3], pears [4], grapes [5], pomegranates [6], loquats [7], and other horticultural crops. Global warming and the increasing prevalence of dwarf apple cultivars (*Malus domestica*) have exacerbated such incidences, leading to a steady rise in fruit sunburn cases each year [3].

Fruit peel changes color obviously under the influence of strong light and high temperature. The color change in burned fruit peel is related to the content of photosynthetic pigment and photoprotective pigment in fruit epidermal cells [8]. Under the condition of high-intensity solar radiation, green and red, related to chlorophyll and anthocyanins, in apple fruits were degraded, while yellow and brown, related to carotenoids and different phenolic compounds, were increased, which was the main reason for the change in fruit surface color caused by sunburn [9]. Compounds such as anthocyanins, flavonoids, and phenols can make apple peel cope with excessive visible light, among which, flavonoids and phenols can effectively protect apples from light damage caused by UV-B radiation [8]. After sunburn, the anthocyanins and chlorophyll are destroyed and the content of phenolic substances increases, but the content is different in different stages of sunburn [10].

Procyanidins (PAs), also known as condensed tannins, are phenolic compounds composed of varying amounts of catechins and epicatechins. They can effectively eliminate superoxide anion free radicals and hydroxyl free radicals, and also participate in the metabolism of phosphoric acid and arachidonic acid and protein phosphorylation, protecting lipids from peroxidation damage. These compounds play a dual role in plant color formation and environmental stress resistance [11]. PAs enhance the antioxidant system by inhibiting polyphenol oxidation, preventing lipid peroxidation in cell membranes, and mitigating enzymatic browning in apples [12]. Incorporating PAs into a chitosan membrane matrix improves the preservation of fruits and vegetables while reducing surface browning during storage [13]. Our preliminary results suggest that apple peel with severe sunburn was found to contain significantly lower PA levels than peel with mild sunburn, suggesting that PAs are key protective compounds against sunburn in apple peel [14].

Anthocyanidin reductase (ANR) is a key enzyme in the PAs biosynthesis pathway [15]. It regulates flavonoid synthesis and anthocyanin accumulation, playing a crucial role in plant pigmentation [16]. The overexpression of *MrANR1* and *MrANR2* enhances PA accumulation, whereas the transient silencing of *ANR* in begonia reduced PA levels and altered leaf coloration [17]. While ANR’s role in pigmentation has been extensively studied, researchers have only recently started paying attention to its function in stress responses. The expression of *GbANR47* is significantly higher in resistant fruit varieties than in sensitive ones, with varying levels of methyl jasmonate (MeJA)- and salicylic acid (SA)-responsive elements identified in their promoter regions [18]. In addition, the decreased expression of *MhANR* in *Arabidopsis*, tobacco, and apple callus reduced tolerance to salt–alkali stress by modulating osmoregulatory substances, chlorophyll content, antioxidant enzyme activity, and stress-related gene expressions [19].

Current research on sunburn has primarily focused on its causes, symptoms, and preventive measures, examining its correlations with factors such as fruit variety, site conditions, cultivation practices, and meteorological influences [1]. However, limited studies have investigated the physiological responses during pericarp burn, and the molecular regulatory mechanisms within the pericarp remain to be fully elucidated. Previous work identified MdMYBR9 as a key transcription factor influencing sunburn formation in apple peel [20], but its specific regulatory pathway remains unclear. This study demonstrates that MdMYBR9 regulates the expression of the downstream structural gene *MdANR*, providing further insight into the molecular mechanisms underlying sunburn resistance in apple peel. These findings offer a theoretical foundation for developing sunburn-resistant apple varieties and improving sunburn prevention strategies.

## 2. Results

### 2.1. Effects of Different Bagging Removal Times on MdANR Expression in Apple Peel

The surface color of the *Ruixianghong* apples varied significantly depending on the duration of bagging. Bag removal at 80 DAF exhibited a yellowish-brown peel, and the peel showed some burns (Figure 1A). As the bagging removal period was postponed, the peel initially turned red before gradually becoming lighter in color (Figure 1A).

Previous findings suggest that *MdMYBR9* enhances the sunburn resistance of apple peel, likely through the regulation of *MdANR*. To further validate the role of *MdANR*, its expression pattern was analyzed across different developmental stages. RT-qPCR was performed to quantify *MdANR* expression in *Ruixianghong* apple peel at various harvesting times (Figure 1B). The results showed a decline in *MdANR* expression as light intensity and temperature decreased between 80 and 140 DAF, indicating that *MdANR* is positively regulated by these environmental factors.

### 2.2. Gene Structure and Phylogenetic Analysis of MdANR

The CDS of *MdANR* was cloned from the peel of *Ruixianghong*, and its gene structure was analyzed, which revealed the presence of two introns (Figure 2A). Conserved domain predictions indicated that *MdANR* belongs to the NADB-Rossmann superfamily (Figure 2B). Tertiary structure analysis shows that the *MdANR* protein consists of multiple single-helical structures (Figure 2C). To further investigate its function, a phylogenetic analysis was performed using *MdANR* and ANR protein sequences from other species (Figure 2D). The results classified the evolutionary tree into three major groups, with *MdANR* clustering with *MsANR-like*, *PbANR*, and *PcANR1*. Studies have shown that the *ANR* gene plays a key role in regulating PA biosynthesis in these species [21]. It is speculated that ANR in apple species has close to a homologous relationship and may have similar functions.

### 2.3. Subcellular Localization of MdANR

To determine the intracellular localization of *MdANR*, a subcellular localization analysis was conducted in tobacco leaves. Agrobacterium GV3101 carrying pCMBIA2300-GFP fusion protein was transformed into tobacco leaves. Fluorescence signals were observed under a laser confocal scanning microscope, generating bright-field, NLS-mCherry, fluorescence, and merged images (Figure 3). Confocal microscopy revealed that the fluorescence from *35S::MdANR-GFP* was uniformly distributed along the cell membrane, indicating that *MdANR* is a membrane-localized protein and that its gene expression products function at the cell membrane.

### 2.4. Functional Verification of MdANR in Apple Peel

To investigate the role of *MdANR* in mitigating sunburn in apple peel, *Agrobacterium* carrying the *MdANR* expression vector was transiently injected into the peel of *Ruixianghong* apples. The injected fruits were kept in darkness for 4 days before being exposed to high-temperature and strong-light treatment for 2 h. Phenotypic observations revealed that apples injected with the *2300-MdANR* vector exhibited the least peel damage, whereas those injected with the *TRV-MdANR* silencing vector showed the most severe sunburn symptoms (Figure 4A). The different treated peel was then collected for further analysis. RT-qPCR indicated that *MdANR* expression in the pericarp differed significantly between the overexpression and silencing groups compared with the control (Figure 4B), confirming the successful transformation of *MdANR* in apple peel. The changes in the levels of the main metabolites in the pericarp following transient *MdANR* expression were further analyzed. Compared with the control and TRV-MdANR silencing vector group, apple peel overexpressing *MdANR* exhibited significantly higher levels of PAs, phloretin, and chlorogenic acid (Figure 4C). This shows that the accumulation of these substances enhances the resistance of apple peel to high temperatures and strong light. These results indicate that *MdANR* overexpression enhances PAs accumulation, which enhances resistance to high-temperature and strong-light stress in apple peel.

### 2.5. Functional Verification of MdANR in Apple Callus

To further validate the function of *MdANR*, stable transformation was performed in apple callus using *Agrobacterium* carrying the *MdANR* overexpression vector. Transgenic apple callus lines with *MdANR* overexpression (OE-1, OE-2, and OE-5) were successfully generated (Figure 5). PCR analysis confirmed the presence of the transgene in the overexpression lines, as indicated by the appearance of white bands in the transgenic samples (Figure 5A). RT-qPCR analysis further demonstrated significantly higher *MdANR* expression levels in transgenic callus compared with the WT control (Figure 5B), confirming stable gene expression. Following high-temperature and strong-light treatment, phenotypic differences were observed between the WT and transgenic calli. WT callus exhibited more pronounced browning than transgenic callus, suggesting that *MdANR* overexpression enhances stress resistance (Figure 5C). Metabolite analysis revealed significant increases in procyanidin (PA), salicylic acid, and chlorogenic acid levels in transgenic callus lines, with concentrations 11.78, 1.37, and 1.96 times higher than those in the WT callus, respectively (Figure 5D). These results indicate that *MdANR* overexpression promotes the accumulation of PAs and other stress-related metabolites in apple callus.

### 2.6. MdMYBR9 Binds to the MdANR Promoter and Induces Its Activity

To determine whether *MdANR* is directly regulated by the upstream transcription factor *MdMYBR9*, a series of molecular assays, including a dual-luciferase reporter assay, yeast one-hybrid (Y1H) assay, and EMSA, were conducted. The *MdANR* promoter was cloned into the pGreen II 0800-LUC vector, whereas the CDS of *MdMYBR9* was inserted into the pGreen II 62-SK vector. When *35S::MdMYBR9* and *proMdANR::LUC* were co-infiltrated into tobacco leaves, LUC activity was significantly elevated, indicating that *MdMYBR9* activated the *MdANR* promoter (Figure 6A). The dual-luciferase assay substantiated these findings, as the LUC/REN ratio demonstrated that *MdMYBR9* significantly upregulated *MdANR* promoter activity (Figure 6B). For the Y1H assay, the *MdANR* promoter was inserted into the pAbAi vector, whereas *MdMYBR9* was cloned into the pGADT7 vector to generate the *MdMYBR9*-AD recombinant construct. This construct was introduced into yeast strains containing *MdANR*-pAbAi. Yeast cells harboring both *MdMYBR9*-AD and *MdANR*-pAbAi grew on selection medium containing 300 ng/mL AbA, whereas those containing *MdANR*-pAbAi alone or the AD empty vector failed to grow, confirming that *MdMYBR9* binds directly to the *MdANR* promoter (Figure 6C). The EMSA results provided further evidence of this interaction. No gel shift was observed when the biotin-labeled probe was added alone. However, after incubation with the *MdMYBR9* protein, a clear shift was detected, indicating binding. The addition of an unlabeled cold probe reduced the intensity of the binding signal, whereas no shift was observed when a mutant probe was used (Figure 6D). Collectively, these results confirm that *MdMYBR9* directly binds to the *MdANR* promoter, activates its transcription, and promotes *MdANR* expression.

## 3. Discussion

*ANR* is a key gene involved in the biosynthesis of procyanidin (PA) monomers and catalyzes the conversion of various anthocyanidin monomers into flavanols in vitro [22]. Under environmental stress, the upregulation of *ANR* expression enhances PA accumulation, playing a crucial role in stress adaptation [23]. The ectopic expression of the apple *MdANR* gene in tobacco has been shown to positively regulate PA biosynthesis. However, transcript levels of *MdANR1* and *MdANR2* gradually decline throughout fruit development, suggesting a developmental regulation of PA biosynthesis [24]. Similarly, the ectopic expression of *MnANR* and *MnLAR* in tobacco suppressed anthocyanin biosynthesis gene expression, leading to reduced anthocyanin content and increased resistance to gray mold. Transgenic tobacco plants exhibited fewer disease symptoms than WT plants, indicating a protective role of *ANR* in stress resistance [25]. In this study, *MdANR* expression declined progressively with decreases in light intensity and temperature during different stages of apple fruit development (Figure 1B). These findings suggest that *MdANR* expression is developmentally regulated and responds positively to high-temperature and strong-light conditions.

Subcellular localization analysis is essential for determining the intracellular distribution of a protein, which provides critical insights into its biological function [26]. To date, the subcellular localization of the apple *ANR* enzyme has not been reported. The present study revealed that *MdANR* is localized to the cell membrane in tobacco cells (Figure 3), suggesting that *MdANR* functions at the membrane level, possibly in stress responses and metabolic regulation.

PAs are potent antioxidants that scavenge free radicals, protecting plants from biotic and abiotic stresses throughout growth and development. Their antioxidant capacity is reported to be 20 times greater than that of vitamin C and 50 times higher than that of vitamin E [27]. The biosynthesis of PAs is directly regulated by anthocyanidin reductase (ANR) and leucoanthocyanidin reductase (LAR) [28]. Light plays a crucial role in PA synthesis, as plant stems and fruits exposed to higher light intensities develop darker or redder pigmentation due to increased anthocyanin and PA accumulation. This process not only enhances coloration, but also protects tissues from excessive light-induced damage [29]. While PAs have been extensively studied in relation to plant pigmentation, their antioxidant properties have been gradually garnering more attention [30]. Red-fleshed apples exhibit significantly slower browning rates than white-fleshed varieties due to their higher PA content, which enhances their antioxidant activity [31]. Under environmental stress conditions, such as intense light and strong ultraviolet radiation, PAs protect plants by neutralizing reactive oxygen species (ROS) [32,33]. In this study, the transient overexpression of *MdANR* in apple peel (Figure 4) and stable genetic transformation in apple callus (Figure 5) led to a significant increase in PA content compared with the control. This enhanced PA accumulation improved the resistance of apple peel and callus to high-temperature and strong-light stress, which confirmed the protective role of *MdANR*.

Extensive research has explored the regulation of *ANR* by transcription factors, including MYB, NAC, and WD40. Studies suggest that *ANR* is directly regulated by MYB transcription factors, whereas other transcription factors exert indirect regulatory effects [34]. In rose petals, *RhMYB1* activates *RhMYB123* by binding to its promoter, and both proteins cooperatively promote PA biosynthesis by binding to the promoters of *ANR* and *LAR* genes [35]. Similarly, in walnut, *JrMYB12* and *JrbHLH42* interact to form a complex that significantly enhances the promoter activity of *JrLAR* and *JrANR* in PA biosynthesis, either independently or in combination [36]. In apple, the *MdERF1B* transcription factor activates *MdMYB*, which, in turn, regulates PA biosynthesis and enhances antioxidant capacity [37]. Under low-temperature stress, *MdMYB23* directly binds to the *MdANR* promoter, activating its expression, promoting PA accumulation, and enhancing ROS scavenging [23]. Additionally, the overexpression of *MdMYB9* and *MdMYB11* in apple callus under MeJA treatment led to the upregulation of *MdANS* and *MdANR* expression, resulting in the increased accumulation of anthocyanins and PAs [38]. In this study, a Y1H assay, dual-luciferase reporter assay, and EMSA collectively demonstrated that *MdMYBR9* binds directly to the *MdANR* promoter and activates its transcription under high-temperature and strong-light-intensity stress (Figure 6); the study is the first to investigate *MdMYBR9-MdANR* interaction under high-temperature and -light stress. Based on these findings, we propose a new regulatory model in which *MdMYBR9* enhances apple peel resistance to sunburn by activating *MdANR* expression, thereby promoting PA biosynthesis. Overall, we report the *MdMYBR9-MdANR-PAs* regulatory pathway as a key mechanism in mitigating apple peel sunburn (Figure 7).

The authors’ previous research demonstrated that the binding of MdGATA15 to the promoter of MdANR could also upregulate the biosynthesis of proanthocyanidins, thereby reducing the degree of sunburn on the apple fruit surface [14]. The GATA family is a type of zinc finger transcription factor. Most studies have mainly focused on regulating the growth and development of plants, while some GATA family members have been proven to be involved in stress responses. In this study, MdMYBR9 was shown to have a direct regulatory effect on MdANR. Whether MdMYBR9 has an interaction relationship with MdGATA15 that jointly promotes the expression of MdANR remains to be further studied.

## 4. Materials and Methods

### 4.1. Experimental Design of Apple Picking Bag Treatment

The experiment was conducted at the Baishui Apple Test Station of Northwest A&F University (35.27 °N, 109.16 °E) at an altitude of 850 m. The site has an annual average temperature of 10.5 °C and a frost-free period of 209 days. The study used 7-year-old bagged Ruixianghong apple trees grafted onto *M26* rootstock, trained in a trunk-shaped form with a planting density of 1.5 m × 4 m and an average tree height of approximately 3.0 m. Five trees with uniform growth were randomly selected, and their unshielded fruits were bagged 80 days after flowering (DAF) (28 June). Bag removal treatments were applied every 15 days, at 80, 95, 110, 125, and 140 DAF. All the processed apples were picked by 140 days, and each treatment group included 20 fruits. The irradiated fruit peel (approximately 2 mm thick) from each treatment was immediately frozen in liquid nitrogen and stored at −80 °C for fluorescence quantitative PCR analysis of *MdANR* expression. Each experiment was performed in triplicate.

### 4.2. Selection of Transgenic Materials

The apple material used for instantaneous transformation in this experiment consisted of bagged *Ruixianghong* apple fruits collected from the Baishui Apple Test Station at 170 DAF. For stable genetic transformation, *Wang Lin* apple callus was used. The callus was cultured in darkness at 25 °C, subcultured every 15 to 20 days, and tested after six rounds of subculturing.

### 4.3. Ligation of Gene Clones and Vectors for Genetic Transformation

The genome information of *Golden Delicious* was retrieved using the genome analysis software UltraEdit (64-bit; version 22.20). The coding sequence (CDS) of the cloned genes was identified, primers were designed accordingly, and the CDS was successfully cloned. The primer sequences used in this study are listed in Table 1.

The overexpression vector pCAMBIA2300 and the silencing vector TRV plasmid were provided by Yang Yazhou from Northwest A&F University. The vectors were digested with restriction enzymes and incubated at 37 °C for 45 min, followed by ligation with the target insert. The ligated product was then incubated at 37 °C for 50 min before being used for *Escherichia coli* transformation.

### 4.4. Transient Transformation of Apple Fruit

The amplified gene product was transformed into the competent cells of *E. coli* DH5α. Plasmids from the correctly sequenced colonies were extracted and subsequently transformed into *Agrobacterium tumefaciens* GV3101 (*Weti*). The *Agrobacterium* culture was resuspended in 300 mL of buffer containing 10 mmol L^−1^ MgCl_2_, 10 mmol L^−1^ 2-(N-morpholino) ethanesulfonic acid (MES), and 150 μmol L^−1^ acetosyringone (AS), adjusted to an optical density of OD_600_ = 1.0, and incubated at room temperature in the dark for 3 h. The prepared suspension was then injected into apple fruits using a 1 mL syringe.

Fruits still attached to the trees were injected in vivo [39]. The central pericarp of each fruit was injected with 1 mL of an *Agrobacterium* suspension containing either the overexpression vector (2300-*MdANR*), the empty vector (2300) as a negative control, or the silencing vector (TRV-*MdANR*). After injection, all the fruits were re-bagged, and eight uniformly sized fruits were selected for each treatment. Four days later, the bagged apples were harvested and transported to the laboratory. The injection site on the fruit peel was then subjected to high-temperature and strong-light treatment at 45 °C under an irradiance of 1200 μmol m^−2^ s^−1^ for 2 h. Following treatment, a 2 mm thick peel section from the injection site was excised using a sterile scalpel for real-time quantitative PCR (RT-qPCR) and metabolite analysis.

### 4.5. Stable Genetic Transformation of Apple Callus

The CDS of *MdANR* was inserted into the pCAMBIA2300 overexpression vector and transformed into *A. tumefaciens* GV3101. The bacterial culture was adjusted to an OD_600_ of 0.8–0.9 and transferred into a conical flask containing wild-type (WT) apple callus. The suspension was incubated at 28 °C with shaking at 200 rpm in the dark for 1 h. The bacterial solution was then filtered through gauze, and excess moisture was removed from the callus using filter paper before plating it onto a selection medium. After 15 days of dark incubation, the callus was harvested for DNA and RNA extraction to confirm the successful transformation.

Both WT and transgenic calli were exposed to the high-temperature and strong-light treatment at 45 °C under an irradiance of 1200 μmol m^−2^ s^−1^ for 2 h. Following treatment, real-time quantitative PCR (RT-qPCR) and metabolite analysis were performed to assess gene expression and biochemical changes.

### 4.6. RT-qPCR Expression Analysis

Total RNA was extracted using the RNAprep Pure Polysaccharide Polyphenol Plant Total RNA Extraction Kit (Tiangen, Beijing, China) following the manufacturer’s instructions. Complementary DNA (cDNA) was synthesized using the EasyScript One-Step cDNA Synthesis SuperMix Kit (TransGen Biotech, Beijing, China). RT-qPCR was performed on an ABI StepOnePlus system (Applied Biosystems, Carlsbad, CA, USA) using the PerfectStart^®^ Green qPCR SuperMix Kit [40]. *MdActin* served as the reference gene. All experiments were conducted in triplicate.

### 4.7. Metabolite Content Determination and Analysis

The flavonoid and phenolic compound contents were analyzed using high-performance liquid chromatography–tandem mass spectrometry (HPLC–MS/MS, CA, USA) [41]. A 0.5 g sample of apple peel was ground into powder and extracted with 2 mL of 80% (*v*/*v*) methanol–water solution. The mixture was vortexed for 10 min, subjected to ultrasonic extraction at 40 °C for 60 min, and centrifuged at 5000 rpm for 10 min. The resulting supernatant was filtered through a 0.22 μm organic phase filter membrane and stored in brown vials. Mass spectrometric analysis was conducted using a QTRAP 5500 ion trap mass spectrometer (AB SCIEX, Framingham, MA, USA) operating in the negative electrospray ionization (ESI) mode. Phenolic compounds were detected using multiple reaction monitoring mode. Chromatographic separation was performed on an InertSustain AQ-C18 column (150 × 4.6 mm, 5 µm; GL Sciences, Torrance, CA, USA) at 35 °C. The sample injection volume was 5 μL and the flow rate was 0.7 mL min^−1^. The mobile phase consisted of 1% formic acid aqueous solution (A) and acetonitrile (B). Compound quantification was based on the peak area ratio of the analyte to the standard curve. All standard compounds were obtained from Shanghai Yuanye Biotechnology Co., Ltd. (Shanghai, China).

### 4.8. Gene Structure Analysis

The CDS, gene sequence, and protein sequence were retrieved using NCBI online tools. The intron–exon structures were analyzed using the Gene Structure Display Server 2.0 platform. A phylogenetic tree was constructed using MEGA (Molecular Evolutionary Genetic Analysis) V7.0 Software. Additionally, NCBI tools were used to predict conserved domains within gene sequences [42].

### 4.9. Subcellular Localization Analysis

Following the method described by Liu [43], the CDS of *MdANR* was inserted into the pCMBIA2300-GFP vector through homologous recombination. The resulting fusion construct, *35S::MdANR-GFP*, was introduced into *Nicotiana benthamiana* leaves via *Agrobacterium tumefaciens* GV3101-mediated transformation. A *35S::GFP* vector was used as a control. For transformation, *Agrobacterium* was injected into the abaxial side of 4-week-old tobacco leaves using a 1 mL syringe. Each tobacco plant received injections in six leaves, with each leaf vein side injected once. Two days after inoculation, images were captured by confocal microscopy (Leica TCS-SP8 SR, Wetzlar, Germany) with a 514 nm laser.

### 4.10. Dual-Luciferase Reporter Assay

Following the method described by Wang [44], the promoter of *MdANR* was cloned into the pGreen II 0800-LUC vector, whereas the CDS of *MdMYBR9* was inserted into the pGreen II 62-SK vector. The recombinant constructs were transformed into *A. tumefaciens* GV3101 (*pSoup-p19*) and cultured with shaking until the bacterial suspension reached an OD_600_ of 0.8–1.0. The cells were then collected by centrifugation and resuspended in a buffer containing 10 mmol L^−1^ MgCl_2_, 10 mmol L^−1^ MES, and 150 μmol L^−1^ AS. The OD_600_ of the final bacterial suspension was adjusted to 0.5, and the prepared suspension was incubated at room temperature in the dark for 3 h before injection. The transformed *Agrobacterium* was injected into tobacco leaves, which were first kept in darkness at room temperature for 12 h and then exposed to light. Three untreated tobacco plants were maintained as controls. After three days, leaf samples from the injection sites were collected, and firefly luciferase (Luc) and *Renilla* luciferase (Ren) activities were measured using an enzyme marker. The Luc/Ren ratio was calculated, and luminescence analysis was performed using the in vivo molecular marker imaging system Lumazone Pylon 2048B (Princeton, NJ, USA).

### 4.11. Yeast One-Hybrid Assay

Following the method described by Zhang [45], the PlantCARE database was used to predict cis-acting elements in the *MdANR* gene promoter. Two binding sites for *MdMYBR9* (CAACCA and TAACTG) were identified. The *MdANR* promoter fragment was cloned into the pAbAi vector through homologous recombination, linearized with the restriction enzyme *BstB1*, and integrated into the *Y1H Gold* yeast strain to generate a bait-specific reporter strain. Simultaneously, the minimum concentration of aureobasidin A (AbA) required to suppress the background expression of the reporter gene in the bait strain was determined. The CDS of *MdMYBR9* was cloned into the pGADT7 (AD) vector to generate the *MdMYBR9*-AD recombinant construct. This construct was then introduced into the yeast strain containing the *MdANR*-pAbAi reporter vector. The *p53*-AbAi vector was used as a positive control, whereas the empty pGADT7 vector served as a negative control. The interaction between *MdMYBR9* and the *MdANR* promoter was assessed based on yeast growth under selective conditions.

### 4.12. Electrophoretic Mobility Shift Assay

Following the method described by Wang [44], the CDS of *MdMYBR9* was cloned into an expression vector and transformed into an *E. coli* strain (DH 5α). Expression of *MdMYBR9*-His fusion protein was induced with 1 mM isopropyl β-D-1-thiogalactopyranoside (IPTG) treatment at 16 °C for 16 h and at 37 °C for 4 h. The protein was purified using a His-tagged protein purification kit (Beyotime, Shanghai, China) according to the manufacturer’s instructions. The *MdANR* probe was synthesized by Aiji Baike Biotechnology Co., Ltd. (Wuhan, China). The electrophoretic mobility shift assay (EMSA) included gel preparation, binding reaction setup, electrophoresis, membrane transfer, hybridization, blocking, washing, and elution. Chemiluminescence detection and imaging were performed to assess protein–DNA interactions.

### 4.13. Data Analysis

A completely randomized design was used for the experiment. Data analysis was performed with the IBM SPSS Statistics 26 statistics program, and Duncan’s new complex range method was used to test the significance differences in the mean values in this study. Plots were generated using Origin 2018 (OriginLab, Northampton, MA, USA) and R Studio version 3.6.1.

## 5. Conclusions

In this study, *MdANR* was cloned from apple, and its structural and physicochemical properties were characterized. Its function was validated through transient transformation in apple peel and stable genetic transformation in apple callus. Additionally, a Y1H assay, dual-luciferase reporter assay, and EMSA demonstrated that *MdMYBR9* binds to the promoter of the downstream structural gene *MdANR*, activating its expression, promoting PA biosynthesis, and enhancing sunburn resistance in apple peel. This study elucidates the molecular regulatory mechanism underlying apple fruit sunburn and provides a theoretical basis for breeding and selecting sunburn-resistant apple varieties.

## Figures and Tables

**Figure 1 ijms-26-04656-f001:**
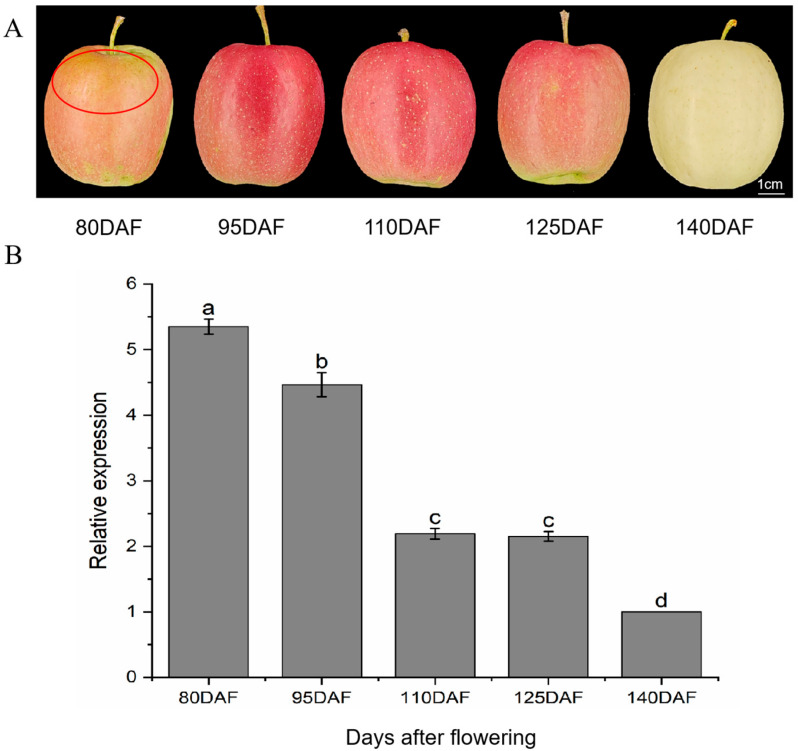
Changes in apple surface appearance and *MdANR* expression levels at different harvesting stages. (**A**) Phenotypic changes in bagged apple fruits at different days after flowering (DAF). (**B**) Expression levels of *MdANR* in pericarp at corresponding time points. Different letters indicate statistically significant differences across indicated DAF (*p* < 0.05).

**Figure 2 ijms-26-04656-f002:**
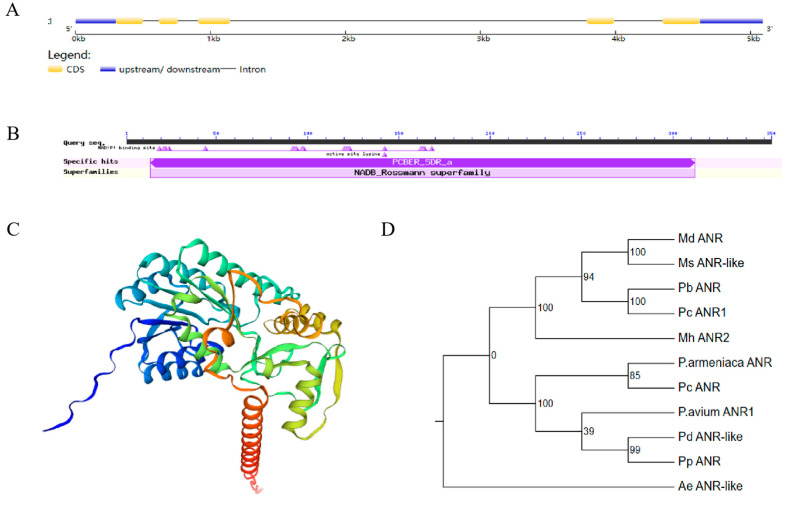
Bioinformatics analysis of *MdANR*. (**A**) Intron–exon structure of *MdANR* gene. (**B**) Predicted conserved domain of *MdANR* protein. (**C**) Predicted secondary structure of *MdANR* protein. (**D**) Phylogenetic tree of *ANR* in *Malus domestica* and other tree species. Sequences included in analysis are from *Malus domestica* (*Md*) *ANR*, *Malus sylvestris* (*Ms*) *ANR-like*, *Pyrus bretschneideri* (*Pb*) *ANR*, *Pyrus communis* (*Pc*) *ANR1*, *Malus hybrid* (*Mh*) *ANR2*, *Prunus avium* (*P. avium*) *ANR1*, *Prunus dulcis* (*Pd*) *ANR-like*, *Prunus persica* (*Pp*) *ANR*, *Prunus armeniaca* (*P. armeniaca*) *ANR*, *Actinidia eriantha* (*Ae*) *ANR-like*, and *Prunus cerasus* (*Pc*) *ANR*.

**Figure 3 ijms-26-04656-f003:**
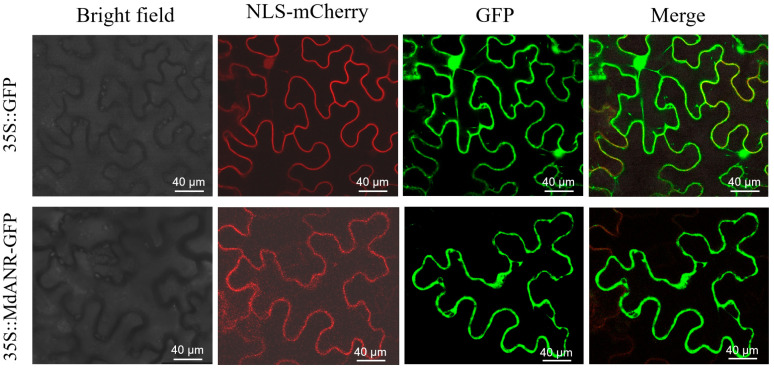
The subcellular localization of *MdANR.* The empty vector (35S::GFP) was used as a control. Nuclear staining was performed using mCherry. GFP represents green fluorescent protein, whereas the merged image shows the overlap of the GFP and mCherry signals, indicating colocalization.

**Figure 4 ijms-26-04656-f004:**
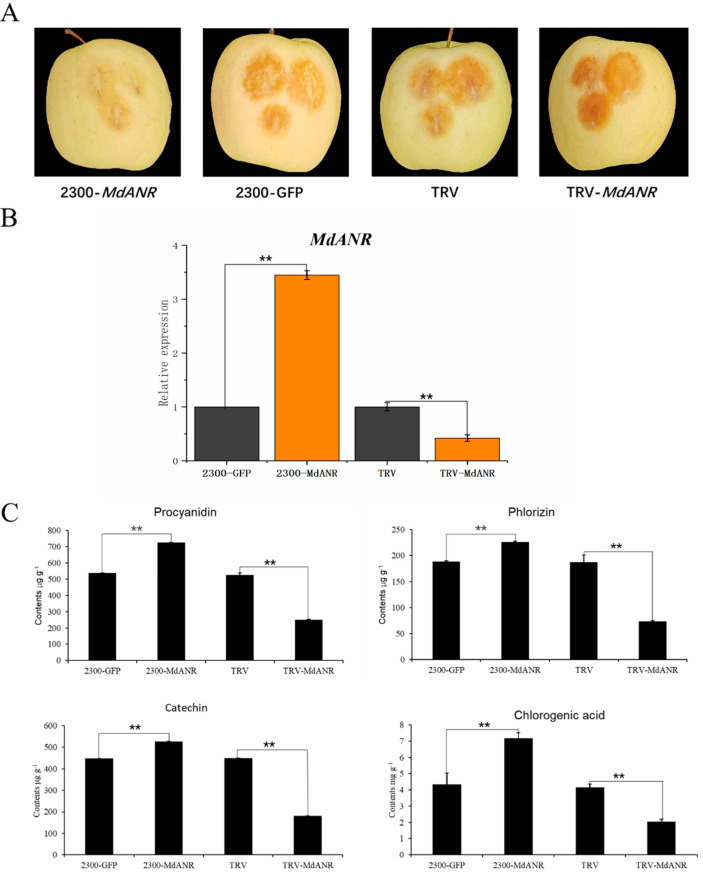
Transient expression of *MdANR* in apple fruit. (**A**) Phenotypic changes in apple peel following transient transformation. (**B**) Relative expression levels of *MdANR* in apple peel. (**C**) Changes in metabolite content in apple peel. Data presented as mean ± standard deviation, and ** indicates statistically significant difference (*p* < 0.01).

**Figure 5 ijms-26-04656-f005:**
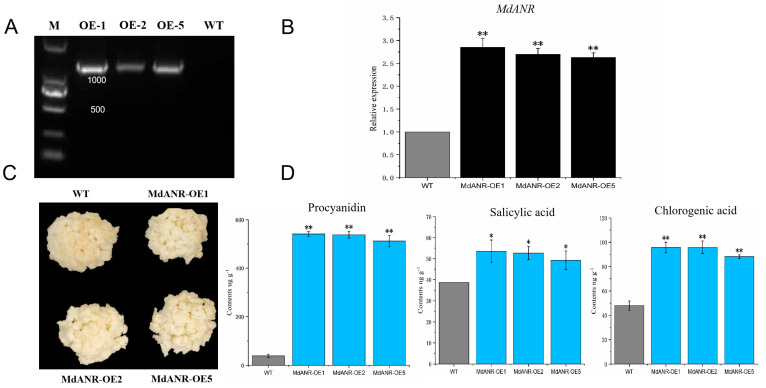
Stable genetic transformation of *MdANR* in apple callus. (**A**) PCR identification of transgenic callus. (**B**) Relative expression levels of *MdANR* in callus. (**C**) Phenotypic differences in callus following transformation. (**D**) Comparison of metabolite content in transgenic and control calli. Error bars represent standard deviation from three biological replicates. Asterisks (*, **) above error bars indicate statistically significant differences (*p* < 0.05 and *p* < 0.01, respectively).

**Figure 6 ijms-26-04656-f006:**
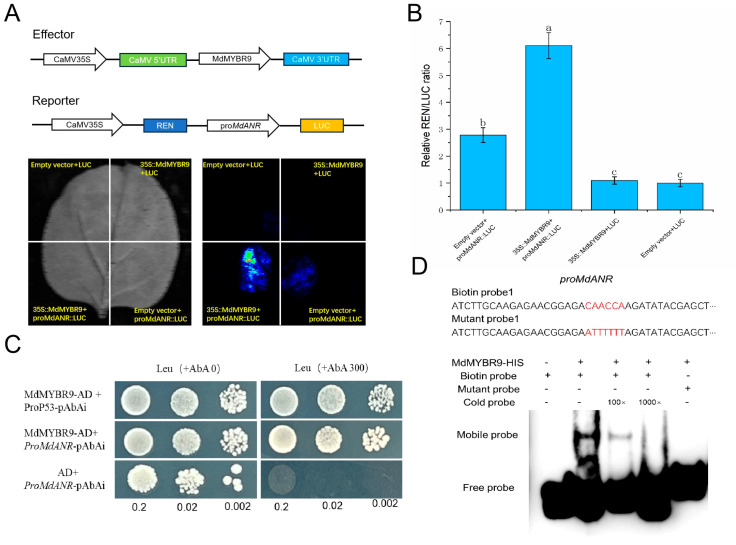
*MdMYBR9* binds to the *MdANR* promoter and induces its expression. *MdMYBR9*–*MdANR* interaction was confirmed using a (**A**) luciferase complementation assay, (**B**) dual-luciferase reporter assay, (**C**) yeast one-hybrid screening assay, (**D**) electrophoretic mobility shift assay. The error bars represent the standard deviation from three biological replicates. The different letters above the error bars indicate statistically significant differences (*p* < 0.05).

**Figure 7 ijms-26-04656-f007:**
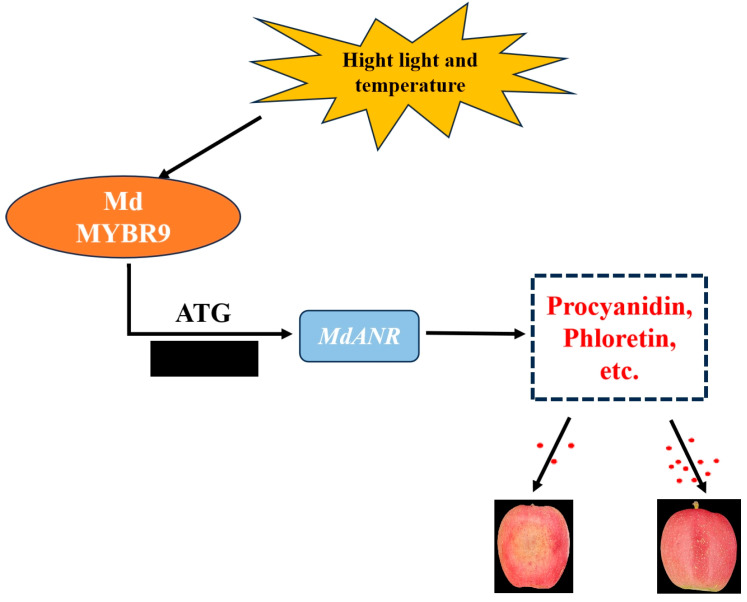
A model of sunburn mediated by MdMYBR9–MdANR interaction. MdMYBR9 promotes the synthesis of PAs and other substances by regulating MdANR expression, the more substances are synthesized, the milder the sunburn.The number of red dots in the figure indicates the amount of substance synthesis.

**Table 1 ijms-26-04656-t001:** Primers used in present study.

Primer Name	Sequence
MdANR-F	ATGACCGTTTCATCTTCTCTTTCTG
MdANR-R	AGCACAAGTGGCAGTGACAGTC
35S-F	TCCTTCGCAAGACCCTTCCTCTAT
Seq-2300-F	CACTTTATGCTTCCGGCTCGTATG
SeqGFP-R	CAGGGTCAGCTTGCCGTAG
2300-XbaI-R	CCATGGTGTCGACTCTAGA
2300-KpnI-F	CGGGGGACGAGCTCGGTACC
MdANR-2300-F	ACGGGGGACGAGCTCGGTACCATGACCGTTTCATCTTCTCTTTCTG
MdANR-2300-R	GGTGTCGACTCTAGAGGATCCAGCACAAGTGGCAGTGACAGTC
MdANR-TRV2-F	GTGAGTAAGGTTACCACATCAATGGGCTTGCATCC
MdANR-TRV2-R	GAGACGCGTGAGCTCAGCACAAGTGGCAGTGACAGTC
MdMYBR9-LUC-F	AGAGGACAGCCCAAGCTGAGCTCATGTCGTCGGGCACGTGC
MdMYBR9-LUC-R	TTTCAGCGTACCGAATTGGTACCGGTGACGCTGATCATGCTATCC
MdANR-LUC-F	CACTATAGGGCGAATTGGGTACCATGGACTTGTTTTTGACCCTCAA
MdANR-LUC-R	TATGTTTTTGGCGTCTTCCATGGGGCTGCTGCTGCTCTTCTTTC
pAbi-F	TCTGTGCTCCTTCCTTCGT
pAbi-R	TGTATTTGTGTTTGCGTGTCTAT
pGADT7-F	AATACCACTACAATGGATGAT
pGADT7-R	ACTGTGCATCGTGCACCATCTC
MdMYBR9- pGADT7-F	GCCATGGAGGCCAGTTGAATTCATGTCGTCGGGCACGTGC
MdMYBR9- pGADT7-R	CAGCTCGAGCTCGATGGATCCTTAGGTGACGCTGATCATGCTATC
MdANR-pAbAi-1-F	ATGAATTGAAAAGCTTATGCCACATCTACCTATGAGACAGAT
MdANR- pAbAi-1-R	GTCGACAGATCCCCGGGTACCCTCCGTTCTCTTGCAAGATAGATCC
MdANR-pAbAi-2-F	ATGAATTGAAAAGCTTAATTAGATGGAGATCTCAACCATAGATATA
MdANR- pAbAi-2-R	GTCGACAGATCCCCGGGTACCCCGGAGCGTCAGTCGGCC
MdANR-pAbAi-3-F	ATGAATTGAAAAGCTTCTAGGAAAACTTTGGAAGAGACTCTAAG
MdANR- pAbAi-3-R	GTCGACAGATCCCCGGGTACCGGCTGCTGCTGCTCTTCTTTC
MdActin-F	TGACCGAATGAGCAAGGAAATTACT
MdActin-R	TACTCAGCTTTGGCAATCCACATC
MdANR-F	GACGGTACGGATATCGGGAAG
MdANR-R	GCAGCAAGGGCTAGTAGGTGA

## Data Availability

No datasets were generated or analyzed during the current study.

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
