# Peer review of "Regulation of MdANR in Anti-Burning Process of Apple Peel"

_ijms, 2025, doi:10.3390/ijms26104656_

Round 1
Reviewer 1 Report
Comments and Suggestions for Authors
Dear
Authors.
The manuscript presents a relevant study on the regulation of MdANR and its impact on sunburn resistance in apples, supported by well-designed experiments such as genetic transformations, dual-luciferase assays, Y1H, and EMSA. However, substantial revisions are recommended to enhance the depth of the discussion by comparing the findings with previous studies on similar mechanisms in other species. Additionally, it is crucial to strengthen the statistical analysis to ensure that the observed differences are supported by robust tests. The conclusion section should also be expanded to more clearly discuss the applicability of these findings in apple breeding programs. Further comments that must be addressed in the revised version of the manuscript are included in the attached file.
Best regards.

Author Response
Dear reviewer, I have modified the manuscript according to your suggestions, please check it,With best wishes

Reviewer 2 Report
Comments and Suggestions for Authors
- The DOI links for several references cannot be accessed, making it impossible to verify the accuracy of the citations.
- The issue of sunburn involves multiple aspects and causal factors. This crucial point has been repeatedly emphasized in the articles cited by the authors. However, this study combines literature on various types of sunburn into one discussion, which may cause confusion regarding mechanisms and lead to deviations in subsequent discussions.
- “Treatments were applied every 15 days, with fruit samples collected at 80, 95, 110, 125, and 140 DAFs. Each treatment group included 20 fruits, and those harvested at 140 DAF served as the control.” What treatment?
- Why was one apple cultivar chosen for sampling, while a different cultivar was selected for the transgenic material?
- This study focuses on sunburn, but the authors did not provide a clear definition or evaluation of sunburn severity. Additionally, differences in sunburn severity are not clearly visible in Figure 1. However, distinct differences in color development are evident. Should the authors clarify whether the primary objective is studying color change or sunburn? These two phenomena differ greatly. Since color change also involves metabolism of anthocyanins and related compounds, can the authors confirm that the findings specifically relate to sunburn?
- The data presented in Figure 4 of this manuscript are nearly identical to those in Figure 5 of the authors' other publication titled "MdGATA15 regulates MdANR to promote the synthesis of procyanidin and enhance the sunburn resistance of apple fruit," with differences appearing only in the scale of values. The proportions are almost exactly the same, which raises curiosity.
- The content of this manuscript and the study titled "MdGATA15 regulates MdANR to promote the synthesis of procyanidin and enhance the sunburn resistance of apple fruit" are highly similar. The timing of publication is nearly identical, and the research methods show minimal differences. I do not consider this manuscript sufficiently novel or significant in terms of scientific advancement. Splitting the research into multiple publications or reusing data for different interpretations raises ethical concerns.
- In addition to the major issues mentioned above, numerous other problems related to writing and method descriptions exist but are not detailed here. I do not recommend further consideration of this manuscript.
Author Response
Dear reviewer
First of all, thank you for your questions and suggestions for this manuscript. I have made corresponding changes in the revised draft submitted again. As for the other manuscript you proposed and mine. "MdGATA15 regulates MdANR to promote the synthesis of procyanidin and enhance the sunburn resistance of apple fruit" Similar questions, I want to explain: The article "MdGATA15" was jointly written by a graduate student and I. When I was studying this topic, I found two transcription factors, MdMYBR9 and MdGATA15, through the joint analysis of transcriptome and metabolome, and found that both of these transcription factors had certain phenotypes through functional verification. Then we found the downstream gene, MdANR. In the initial design of the experiment, I studied MdMYBR9 transcription factor, while my graduate students studied MdGATA15 transcription factor, which proved through experiments that both transcription factors interact with the downstream gene MdANR to produce functions. As for "The proportions are almost exactly the same, which raises curiosity." you mentioned, both of these transcription factors are positively regulating MdANR, and the metabolite trends are inevitably similar. I think this is not the same study, it is very possible that MdMYBR9 and MdGATA15 also have an interaction relationship, which is a follow-up research work.
With best wishes

Round 2
Reviewer 1 Report
Comments and Suggestions for Authors
Dear Author
The revised manuscript shows partial responsiveness to my suggestions but lacks substantial integration in several critical areas. While the abstract was slightly reorganized to emphasize MdANR’s role in sunburn resistance, redundancies and general statements such as “the precise molecular mechanism remains unclear” persist, even though the study claims to address this point. The authors also did not revise the keywords, which continue to repeat terms already in the title, missing an opportunity to improve indexing and scientific visibility.
Despite minor adjustments in the introduction, the logical flow of ideas remains disjointed. Redundancy in the description of procyanidins (PAs) and ANR has not been sufficiently reduced, and the specific contribution of the study to apple breeding or stress-resilient cultivar development is still not clearly articulated. Moreover, the reviewer’s suggestions to improve methodological coherence, especially regarding the unification of transformation approaches and the use of additional controls and replicates, were only marginally addressed. The interpretation of results remains largely descriptive, and deeper insight into the role of MdANR, its protein localization, and the functional impact of individual metabolites is lacking.
Importantly, the revised version still omits a dedicated section on the limitations of the study and potential applications of the findings, which is essential for scientific transparency and relevance. The conclusion remains weak, restating prior findings without offering a critical synthesis or projecting future directions. Overall, while minor textual edits were incorporated, the manuscript fails to fully address the core conceptual and structural recommendations, and its scientific impact remains limited unless these critical areas are strengthened.
Bets regards
Author Response
Dear reviewer, first of all, thank you very much for your evaluation and suggestions. This paper introduces the relevant studies on MdANR regulation and its effects on apple sun resistance. However, there are few relevant studies at home and abroad. Although there are some studies on sunburn, most of them are studies on physiological reactions and related preventive measures. The studies are not in-depth, and no research results on similar mechanisms have been seen, so comparison cannot be made. In addition to the enhanced statistical analysis mentioned by the reviewers, this study has carried out a reasonable difference analysis on the data through Duncan's multiple comparison method, which can ensure that the observed differences are supported by reliable experimental results. Therefore, I don't think it is necessary to revise according to the reviewer's suggestions.
Best regards.
Reviewer 2 Report
Comments and Suggestions for Authors
- The authors did not respond to any of the issues raised during the previous review, and most of the manuscript revisions were not correspondingly adjusted. Firstly, this already does not conform to the standard norms of the journal regarding response to reviewers.
- Secondly, regarding the revised content—for example, “Five trees with uniform growth were randomly selected, and their unshielded fruits were bagged 80 days after flowering (DAF) (June 28). Bag removal treatments were applied every 15 days, with 80, 95, 110, 125, and 140 DAFs. Each treatment group included 20 fruits, and those harvested at 140 DAF served.”—it remains unclear whether each treatment involved 20 fruits per tree or 20 fruits in total. In other words, even after revision, the manuscript still has not reached an acceptable standard.
- Regarding the significant issue of academic ethics, the authors explained that two publications were derived from the same experiment, differing only in their investigation of different transcription factors and genes. This clearly represents a case of academic salami slicing. The editors should independently determine whether this manuscript meets the journal’s academic ethical standards. If ethical concerns exist, further review should not be necessary. If the manuscript is deemed ethically acceptable, editors should take responsibility for any subsequent review.
- The previous review explicitly indicated that, due to concerns regarding academic ethics and other significant issues, not every sentence was thoroughly reviewed. However, the current revision not only fails to respond to previously raised concerns but also neglects to proactively identify and correct additional issues requiring revision. Consequently, the manuscript does not meet the requirements of this journal.
Author Response
Dear reviewer First of all, thank you for your questions and suggestions on this article. Regarding the reviewer's suggestion that "it is not clear whether each treatment method is 20 fruits per tree or 20 fruits in total". I think it is clear in the text that there are 5 treatments, each treating 20 fruits. As for your reference to my other manuscript, "MdGATA15 regulates MdANR to promote the synthesis of procyanidin and enhance the sunburn resistance of apple fruit" which is similar to this study, I would like to explain: The article "MdGATA15" was co-written by one of my graduate students and I. We previously found that MdMYBR9 and MdGATA15 transcription factors both have certain resistance to sunburn of apple peel, and through functional verification, we found that these two transcription factors both have certain phenotypes. In the initial design of the experiment, I studied the MdMYBR9 transcription factor, and my graduate students studied the MdGATA15 transcription factor, and experimentally demonstrated that both transcription factors interact with the downstream gene MdANR to produce function. I think this is not the same study, and it is very possible that MdMYBR9 and MdGATA15 also interact, which needs to be proved by further studies. Therefore, I don't think it is necessary to revise according to the reviewer's suggestions. Best wishes